# To FP8 and Back Again: Quantifying Reduced Precision Effects on LLM Training Stability

## Abstract

The massive computational costs associated with large language model (LLM) pretraining have spurred great interest in reduced-precision floating-point representations to accelerate the process. As a result, the BrainFloat16 (BF16) precision has become the de facto standard for LLM training, with hardware support included in recent generations of accelerators. This trend has gone even further in the latest processors, where FP8 has recently been introduced. However, prior experience with FP16, which was found to be less stable than BF16, raises concerns as to whether FP8, with even fewer bits than FP16, can be a cost-effective option for LLM training. We argue that reduced-precision training schemes must have similar training stability and hyperparameter sensitivities to their higher-precision counterparts in order to be cost-effective. However, we find that currently available methods for FP8 training are not robust enough to allow their use as economical replacements. This prompts us to investigate the stability of reduced-precision LLM training in terms of robustness across random seeds, learning rates, and datasets. To this end, we propose new evaluation techniques and a new metric for quantifying loss landscape sharpness in autoregressive language models. By simulating incremental bit reductions in floating-point representations, we analyze the relationship between representational power and training stability with the intent of aiding future research into the field.

## 1 Introduction

Conversational large language models (LLMs), such as ChatGPT (OpenAI, 2024), Gemini (Team, 2024a;b), Claude (Anthropic, 2024), and HyperCLOVA (Yoo et al., 2024), have captured the imagination of both academics and the public at large with their ability to communicate fluently with humans in natural language. However, these models require unprecedented amounts of computation to train, which has engendered interest in methods to improve their training efficiency.

A popular method of improving computational performance is to reduce the bit count of the floating-point representations used for training (Wang et al., 2018; Sun et al., 2020; Peng et al., 2023). Because reading memory is the main bottleneck in modern processors, a problem known as the "memory wall" (Wulf & McKee, 1995; Kim et al., 2023b), reducing the number of bits that each floating-point number uses can accelerate the computation in proportion to the amount of memory reduced. For example, in processors that support it, computations in BrainFloat16 (BF16) (Kalamkar et al., 2019) can have double the maximum throughput of single precision FP32. Furthermore, the FP32 data type, the highest precision data type used in deep learning, has only half the bits of FP64, the most widely used floating-point data type in scientific computing. The current best practice for LLM training is to use BF16 for most of the LLM training computation, with some sensitive portions, such as layer normalization (Ba et al., 2016), carried out in FP32.

As a natural extension of this development, 8-bit floating-point (FP8) (Wang et al., 2018; Sun et al., 2019; Micikevicius et al., 2022; Peng et al., 2023) and even 4-bit floating-point (Sun et al., 2020) data formats have been proposed to accelerate training even further. However, the naïve application of FP8 to LLM training is unstable and requires additional techniques to become viable. While several methods have been proposed to stabilize training LLMs with FP8, relatively little attention has been paid to quantifying the decrease in stability compared to mixed-precision BF16 training.

Cost reduction is the motivation behind the use of FP8 and other reduced-precision training schemes. Therefore, our concern is not whether LLM pretraining with FP8 is possible but whether it is profitable. For cost savings to be realized, the time per training step must be reduced while the number of training steps is kept to a similar number. Training stability is thus a crucial factor for cost-effective LLM training, considering that additional hyperparameter searches and restarts from training failures can outweigh any gains from raw compute acceleration.

For a rough approximation of the cost, a single p5.48xlarge EC2 instance with 8 H100 GPUs costs USD 98.32 per hour as of the time of writing. On a cluster with 1,024 nodes, this would imply that that spending 20 minutes to restart from a checkpoint saved 40 minutes before the loss spike would cost approximately USD 100K. Therefore, for the newly proposed reduced-precision training schemes to be economical, the models trained on them must be similarly robust to hyperparameter choice and stochastic noise as models trained using higher precision.

Previous experience with training LLMs in FP16 raises further concerns. Teams that have trained LLMs have found that even when gradient scaling and other techniques are applied, the FP16 data type, which has five exponent bits, is much less stable for LLM training than BF16, which has eight exponent bits as in FP32. This raises doubts as to whether FP8, which has even fewer bits than FP16, is a practical option for real-world LLM training.

We motivate our line of inquiry with some surprising findings from experiments on the nanoGPT (Karpathy, 2022) codebase, an open-source implementation of GPT-2 pretraining, where we found that even the current best practice of mixed-precision BF16 can introduce training instabilities. When we compared BF16 and TensorFloat32 (TF32) runs, where we ran training for 5% of the original configuration, we found that the BF16 models diverged for 18 of 188 runs, or approximately 10% of all cases, despite using the same configurations as the default run. In contrast, no cases of loss divergence were found for the 70 TF32 models trained using different random seeds. We compare against the TF32 data type because NVIDIA GPUs do not offer tensor cores in FP32.

This is a surprising finding in light of the fact that most recent LLMs are trained with mixed precision BF16 without a comparison with training on TF32, which has three additional mantissa bits. However, a loss divergence rate of approximately 10% at only 5% of training indicates that even standard BF16 mixed-precision training may add non-trivial instability. If even mixed-precision BF16 can cause instabilities, the effects of using even fewer bits should be investigated further.

We make the following contributions in our work.

- We analyze the hidden training instabilities that emerge from reducing precision by clipping mantissa bits to simulate intermediate-bit representations of floating-point numbers. With these experiments, we find greater instability in the model when exposed to higher learning rates or "dirtier" data.

- We propose a metric for quantifying the loss landscape sharpness of models that can predict when training divergence will occur. As even the removal of mantissa bits has a destabilizing effect on LLM training, we use our metric to predict loss divergences even when the loss curve itself has not yet diverged.

## 2 RELATED WORK

### 2.1 TRAINING STABILITY

Analyzing training instability in LLM pretraining directly is impractical due to the massive costs involved. Instead, smaller language models must be used as proxies. Wortsman et al. (2024) explore the robustness of smaller language models as a proxy for LLM pretraining instability. They find that small models using a larger learning rate show similar instability patterns as larger models, such as the growth of attention layer logits and the divergence of the output logits from the log probabilities. They also explore both the causes of numerical instability in LLM training and mitigating strategies such as applying query/key (QK) normalization (Dehghani et al., 2023).

Keskar et al. (2017) explore the sharpness of loss landscapes during large-batch neural network training, finding that larger batch sizes prevent the model from reaching flat regions of the loss

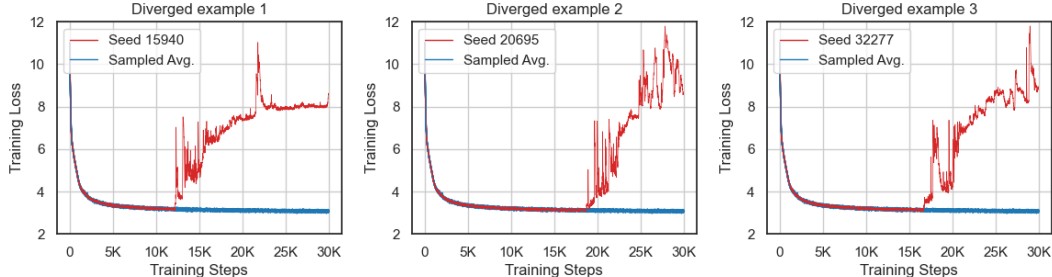

Figure 1: We show three cases of loss divergence on nanoGPT when using the same configurations as the default run except for the random seed. The blue lines indicate the average losses obtained for eight training runs that did not diverge. Of the 188 random seeds that were tested, 18 were found to diverge. As full pretraining requires over 4 days on a single node with 8 A100 GPUs, even for BF16, we perform early stopping at 30K steps, or 5% of the original training steps, requiring approximately 4 hours for a BF16 run and 8 hours for a TF32 run per A100 node with 8 GPUs. Because we only run 5% of the original training, we suspect that the measured divergence rate of approximately 10% underestimates the true rate of training loss divergence.

landscape and causing performance degradation due to the inability to escape local minima. Of most significance to our work, they propose a metric for calculating loss landscape sharpness, which we adapt for LLMs as a proxy for training instability.

Fishman et al. (2024) go further, discovering that Llama 7B models trained in FP8 begin to diverge after 200B tokens of training, supporting our claim that reduced-precision methods induce hidden instabilities that may emerge only later in training. They identify SwiGLU (Shazeer, 2020) as a source of massive activations (Sun et al., 2024) and applying dynamically scaling to reduce the instability. However, it is still unclear whether the proposed FP8 training method is equivalent to BF16 training or is simply stable enough for the experiments involved.

## 2.2 Reduced-precision processors

To improve throughput on computationally intensive matrix multiplication tasks, recently developed processors have been equipped with specialized hardware units such as systolic arrays for TPUs (Jouppi et al., 2017) and tensor cores (NVIDIA, 2020), which serve a similar purpose, for NVIDIA GPUs. These processors can improve throughput by an order of magnitude. For example, on the H100, the peak dense BF16 matrix multiplication throughput on tensor cores is 989.4 TFLOPS, compared to 133.8 TFLOPS when using CUDA cores (NVIDIA, 2022).

However, the number of multiplexer circuits required for the barrel shifter of an $n$-bit floating-point unit is $n \log_2 n$ (Kroening & Strichman, 2008), which incentivizes using smaller floating-point representations. As a result, many mixed-precision techniques perform computationally intensive matrix multiplication in BF16 while preserving sensitive portions of the model, such as the weights and residual path activations, in FP32. Alternatively, Henry et al. (2019) developed a technique to approximate FP32 matrix multiplication using only BF16 by representing a single FP32 value as three BF16 values to accelerate FP32 matrix multiplication without requiring FP32 circuits.

## 2.3 Hybrid FP8

The adoption of the hybrid E5M2/E4M3 formats for neural networks (Micikevicius et al., 2022) in recent generations of processors, such as the NVIDIA H100 and the Intel Gaudi v2, has spurred interest in stable FP8 training. The hybrid FP8 format, where E4M3 is used for the forward pass for its greater resolution, and E5M2 is used for the backward pass for its greater range, was first proposed by Wang et al. (2018) as a means to accelerate neural networks.

Sun et al. (2019) built on this work to propose various techniques for stabilizing training, such as stochastic rounding, chunk-based accumulation, and Kahan summation during the optimizer update. However, the number of techniques that can be used in practice is limited by whether the technique in

Figure 2: Loss landscape diagrams for Llama 120M E8M3 at 5K steps (left) and 10K steps (right). Even during loss divergence, the loss landscape visualized using the method in (Li et al., 2018) appears smooth, motivating our introduction of a new loss landscape sharpness metric. The value of the validation loss is included at each point of the loss landscape.

question can be applied without slowing the computation. As currently available NVIDIA GPUs, by far the accelerators with the greatest adoption, do not support these techniques natively, the overhead caused by the software-based implementations cancels out any gains from the reduced precision.

## 2.4 MS-AMP

Introduced in Peng et al. (2023), MS-AMP is an automatic mixed-precision package to utilize FP8 that offers multiple optimization levels to allow for differing model sensitivities when applying reduced precision to the computations and communications of neural network training. Our experiments use the O1 optimization level of MS-AMP, which performs GEMM computations in FP8 while reducing the weights to half-precision and uses FP8 states for the weight gradients and all-reduce communications. MS-AMP offers additional optimizations for the optimizer buffer in level O2 optimization and for distributed communication in level O3 optimization, but we use only the most basic optimization scheme so as to verify the effects of the least invasive modifications.

## 3 METHODS

We seek to answer whether sub-BF16 training is competitive with standard mixed-precision BF16 training from a cost-effectiveness point of view. To be cost-effective, reduced-precision training schemes must have minimal increases in training instability and changes to hyperparameters. To better analyze the effect of reduced precision on training stability, we aim to quantify the effects of gradually reducing floating-point representations bit by bit for both the exponent and the mantissa. Hopefully, analyzing the intermediate bit representations will better illuminate the interaction between bit width and training stability.

Our intermediate-bit floating-point experiments use the TinyLlama (Zhang et al., 2024) repository, which implements the widely used Llama (Touvron et al., 2023a;b) architecture. TinyLlama is an open-source implementation of Llama pretraining that uses a mix of the SlimPajama (Soboleva et al., 2023) and StarCoder (Li et al., 2023) datasets. It also includes performance optimizations such as Flash Attention v2 (Dao, 2023). We use the default learning rate of $lr = 4e - 4$, global batch size 512, and the same learning rate schedule as in the original code. The 120M models use a sequence length of 2048, while the 7B models use a sequence length of 4096.

## 3.1 SHARPNESS METRIC

To better investigate the model state when loss divergence occurs, we attempted to visualize the loss landscape of the Llama models with the method proposed by Li et al. (2018). However, as shown in Figure 2, we found that even when the model is clearly in the process of loss divergence, the generated visualizations remain smooth. We emphasize that we do not seek the loss landscape sharpness per se, but a proxy to provide a quantitative measurement of the underlying training instability.

Figure 3: Diagram showing the precisions used in a Llama decoder block (best seen in color). The activations in the path of the residual connection are kept in FP32, as are the model weights and embeddings. The LayerNorm and RoPE layers use FP32 internally for their computations. The Flash Attention kernel uses BF16 with no reduction in precision. All other layers use reduced-precision matrix multiplication that emulates low-precision computation with a high-precision accumulator.

Because of this, we propose an alternative loss landscape sharpness metric that is more suitable for autoregressive models, based on the one proposed in Keskar et al. (2017). We empirically confirm that it is a useful indicator of training instability in the following sections. The main difference between the original metric and our version is that we use the logit of the last token instead of the model input for the calculation. This is because adding noise to the embeddings in a language model has different implications compared to adding noise to input images in a vision model.

Instead of searching the input space as in Keskar et al. (2017), we apply the search algorithm to the logit space of the last token. Searching the logit space has the additional advantage that the forward pass of the model need only be performed once for each measurement, significantly reducing the computational cost. The logit of the last token was chosen because it is not computationally feasible to optimize for the entire output space. Also, due to the autoregressive character of decoder-only Transformer models (Vaswani et al., 2017; Brown et al., 2020), the last token is the only one to receive inputs from all other tokens. In addition, we do not apply random projection as in Keskar et al. (2017) to reduce the stochasticity of the measurement.

**Definition**   Let $y \in \mathbb{R}^{s \times v}$ be the output logit for an autoregressive model of sequence length $s$ and vocabulary size $v$. Then, for $y_i$, the output logit at sequence position $i \in \{1, 2, ..., s\}$, and one-vector $\mathbf{1}_v \in \mathbb{R}^v$, we define a constraint set $\mathcal{C}_\epsilon$ at $i = s$ such that

$$\mathcal{C}_\epsilon \in \{z_s \in \mathbb{R}^v : -\epsilon(|y_s| + \mathbf{1}_v) \leq z_s \leq \epsilon(|y_s| + \mathbf{1}_v)\}. \quad (1)$$

Given $y_s \in \mathbb{R}^v, \epsilon > 0$, and noise vector $z_s$, the loss landscape sharpness $\phi_\epsilon$ for loss function $f$ can be defined as

$$\phi_\epsilon := \frac{max_{z_s \in \mathcal{C}_\epsilon} f(y_s + z_s) - f(y_s)}{1 + f(y_s)} \times 100. \quad (2)$$

The proposed metric can best be thought of as the relative magnitude of the largest loss spike on the logit within the provided bounds. The bounds are set to be the logit magnitudes plus one multiplied by $\epsilon$. The largest spike in the vicinity of the logits is found using the L-BFGS-B algorithm (Liu & Nocedal, 1989), using the SciPy (Virtanen et al., 2020) implementation with the output logit set as the starting point of the search. We set $\epsilon = 5e-4$ for all our experiments following Keskar et al. (2017). However, a hyperparameter sweep on $\epsilon$ shows that the general trend is unaffected by the value of $\epsilon$, only the sharpness value magnitudes. The results are shown in Table 2 of the Appendix.

## 3.2   MASKING

Our experiments use a simplified method of reducing floating-point precision to achieve reasonable throughput. To simulate removing exponent bits, we threshold the values to the minimum and maximum absolute values possible with the given number of exponent bits. Figure 5 depicts the exponent masking process. A bitmask is applied to remove the unrepresentable mantissa bits. The

```
def forward(x, w):
    # x: input, w: weight, out: output
    save_for_backward(x, w)
    masked_x = reduce_precision(x)
    masked_w = reduce_precision(w)
    out = F.linear(masked_x, masked_w)
    masked_out = reduce_precision(out)
    return masked_out
```

Figure 4: PyTorch-like pseudocode for the forward pass.

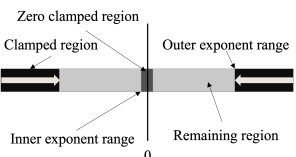

Figure 5: Exponent masking implemented by clamping values that cannot be expressed with the allowed number of exponent bits.

resulting method is an imperfect approximation of reducing the bit count of floating-point numbers. However, it has the advantage of being fast, causing at most a doubling of the time per training step.

Reduced precision operations are applied only on the matrix multiplication computations of the model, excluding the attention computation, which uses the Flash Attention v2 kernel. Following existing FP8 libraries such as TransformerEngine (NVIDIA, 2023), we separate the effects of reducing the computation's precision from reducing the data's precision in storage. As a result, the activations and model weights are kept at their original precision while the inputs and outputs of matrix multiplication are dynamically masked to emulate reduced precision computation with a high-precision accumulator. All states are kept in their original precision, and all operations other than matrix multiplication are performed in their original precision. In Figure 3, we include a diagram indicating the precision of the states and computations in a Llama decoder block.

## 4 RESULTS

### 4.1 MS-AMP EXPERIMENTS

We first analyze the effect of real-world FP8 training by applying the MS-AMP (Peng et al., 2023) library (version 0.4.0) to the nanoGPT codebase. We run all experiments on an H100 node with 8 GPUs to ensure hardware availability of FP8. However, as shown in Figure 6, despite only using the O1 optimization level for MS-AMP, the resulting models show non-trivial performance degradations, especially when the LM head is not excluded from the quantization.

In addition, we also check if the data quality has an effect. In Figure 12, we show the results of using a sample of the FineWeb Edu (Penedo et al., 2024) dataset, which was curated much more rigorously than the OpenWebText (Gokaslan & Cohen, 2019) dataset used in the nanoGPT repository.

These results indicate that the FP8 training scheme in MS-AMP may not converge to the same loss as BF16 training or requires more training steps, depending on factors such as data quality. This strengthens our case that FP8 training may introduce hidden instabilities that are not evident until stress tested against circumstances that were not considered in the original works proposing them.

### 4.2 BIT REDUCTION EXPERIMENTS

We first attempt to identify the points where training instability becomes visible. The emulated reduced-precision representations are denoted using the number of exponent and explicit mantissa bits used. For example, standard BF16 is referred to as E8M7, while a floating-point number with its exponent clamped to seven bits and mantissa clamped to six bits is referred to as E7M6.

We find that removing even a single exponent bit prevents training altogether, resulting in the model failing to progress with any learning using E7M7, confirming previous findings (Henry et al., 2019) that neural network training is more sensitive to exponent bits than mantissa bits. To analyze the cause, we conduct an ablation on the clamping mechanism by either removing only the inner or outer exponent range, as depicted in Figure 5. We find that models with only their inner exponent ranges clamped train normally while models with only their outer exponent ranges clamped do not, indicating that the inability to represent large values is the cause of failure for E7M7. We therefore investigate the effects of removing mantissa bits for the remainder of our experiments.

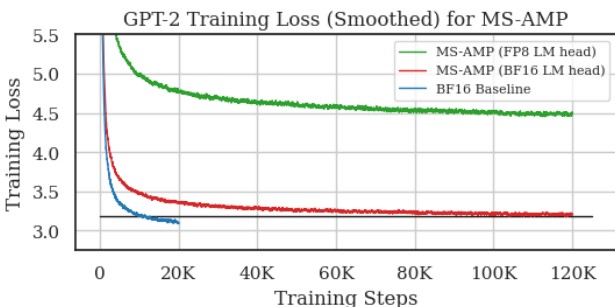

Figure 6: Training losses for GPT-2 training are compared using exponential moving averages to better visualize the general trends. The blue curve indicates the training loss for the baseline BF16 training, while the red curve indicates MS-AMP level O1 training with the LM head excluded from FP8 quantization. The green curve shows the training loss for when the LM head included in FP8 quantization. Not excluding the LM head from FP8 quantization causes a large performance degradation. However, as indicated by the black horizontal line, even when the LM head is excluded, the MS-AMP results do not converge with the BF16 training, even after 120K steps. This result strengthens our case that FP8 pretraining narrows the hyperparameter space where training is stable.

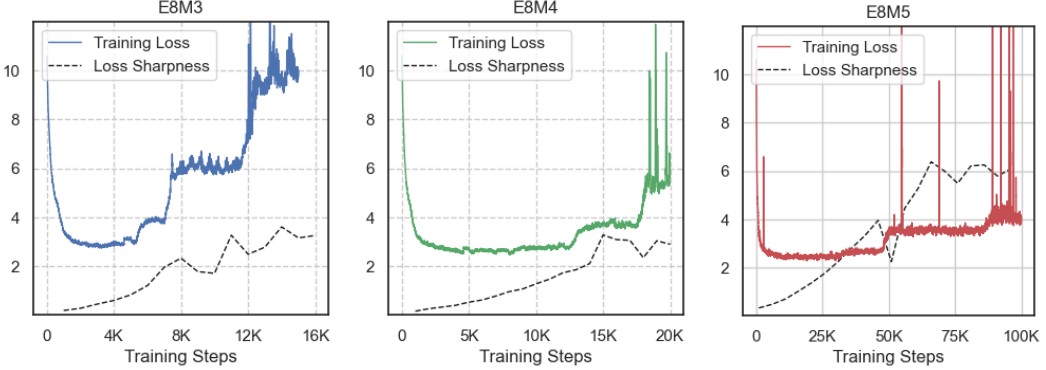

Figure 7: TinyLlama 120M models trained until loss divergence. E8M3, E8M4, and E8M5 models trained for 16K, 20K, and 100K steps, respectively. The dotted black line in each figure indicates the loss landscape sharpness of the model. While no exact sharpness threshold exists for training collapse, a similar pattern is observable across the three precision levels at different training steps.

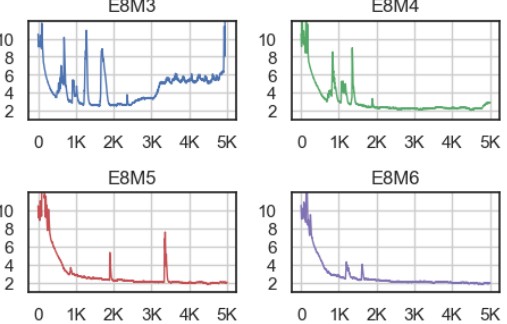

Table 1: Loss landscape sharpness values at $\epsilon = 5e-4$ for Llama v2 7B models trained with TinyLlama for 5,000 steps in Figure 8. Training used a global batch size of 512 and a sequence length of 4096.

| Steps | E8M3 | E8M4 | E8M5 | E8M6 | E8M7 |
|---|---|---|---|---|---|
| 1K | 0.209 | 0.205 | 0.191 | 0.191 | 0.192 |
| 2K | 0.488 | 0.363 | 0.265 | 0.221 | 0.200 |
| 3K | 1.306 | 0.734 | 0.352 | 0.229 | 0.200 |
| 4K | 2.006 | 1.125 | 0.475 | 0.237 | 0.207 |
| 5K | 1.927 | 1.439 | 0.628 | 0.248 | 0.215 |

Figure 8: Llama 7B model training loss curves for different mantissa bits. The x-axis shows training steps, while the y-axis shows the training loss.

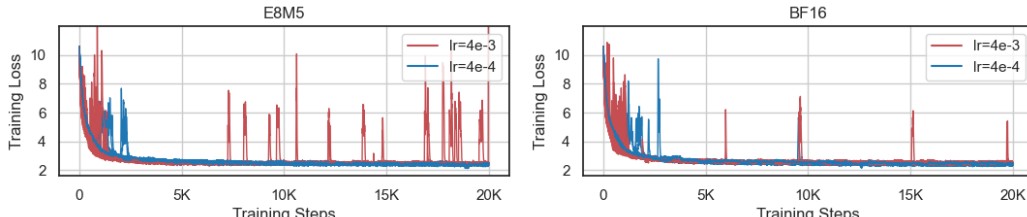

Figure 9: Comparison between Llama 120M models trained using E8M5 masked training (left) and standard BF16 training (right) for $lr = 4e{-}4$ (the default learning rate) and $lr = 4e{-}3$. Using 18 random seeds per configuration, the E8M5 runs show more frequent loss spikes, especially at the higher learning rate, indicating greater training instability.

### 4.3 LOSS LANDSCAPE SHARPNESS

To further uncover the relationship between bit width and training robustness, we use Equation 2 to quantify the degree of training instability increase by measuring the loss landscape instability of Llama models. In Figure 7, we show Llama 120M models trained until their training losses diverge, as well as plotting the loss landscape sharpness values of the models in E8M3, E8M4, and E8M5. Although the points of divergence are different for each model, we can see a general trend of increasing sharpness until the model diverges sharply, after which it cannot revert to its original training trajectory. This pattern is visible despite the large differences in training steps for the three different precisions.

To verify that similar behavior occurs in larger models, we compare the training losses of Llama v2 models with 7B parameters trained for 5,000 steps in Figure 8 and show the measured sharpness values for $\epsilon = 5e{-}4$ in Table 1. Results for other $\epsilon$ values are included in the Appendix and show a similar pattern. We apply early stopping at 5,000 training steps because training a Llama 7B model for 5K steps requires approximately one week on a single node with 8 A100 GPUs. These experiments show that loss divergence is visible in the E8M3 and E8M4 models, while it has yet to emerge in the E8M5 model. However, from Table 1, we can see that the loss landscape sharpness continues to increase for the E8M5 model, even though no signs of instability are yet visible.

The E8M3 and E8M4 models show much higher sharpness values, and both diverge early in training. In contrast, there is only a gradual increase in the loss-landscape sharpness for the E8M7 runs. Figures 7 and 8 show that models gradually increase in sharpness until a threshold level is reached. However, the exact threshold may differ depending on the configurations. These results suggest that models with fewer mantissa bits enter regions of ever greater instability during training, even when these instabilities are not visible in the loss curve. We believe that, in the future, such analysis of loss landscape sharpness can be used to identify when the model is at risk of training loss divergence.

### 4.4 ROBUSTNESS TO LEARNING RATE CHANGES

We further attempt to identify hidden instability in E8M5, which did not diverge during the initial training stages in Figure 7. Inspired by Wortsman et al. (2024), we analyze the robustness of Llama 120M models to changes in the learning rate by comparing training at BF16 with that for E8M5. As seen in Figure 9, the E8M5 training runs have more frequent loss spikes during training, especially when the learning rate is increased to $4e{-}3$. Although no cases of loss divergence were found, we believe that the higher frequency of loss spikes indicates greater sharpness of the loss landscape, supporting our claim that training is more unstable for E8M5 even before loss divergence occurs.

## 5 DISCUSSION

This work proposes quantitative evaluations and analyses of training instabilities when reducing floating-point precision. Our experiments have shown that approximating the reduction of floating-point precision in matrix multiplication destabilizes LLM training and that existing mechanisms to stabilize FP8 training do not offer sufficient robustness to allow their cost-effective use. The issue is

not that FP8 training is not viable. Indeed, we have observed stable training using FP8 using the MS-AMP library in Section 4.1. The issue is that FP8 training causes a narrowing of the hyperparameter space where LLM training can occur stably and with equivalent performance as mixed-precision BF16 training. Because of this narrower hyperparameter space, more resources must be expended on identifying and honing techniques for preventing training collapse. Worse, there is simply no way of knowing if the FP8 training is performing competitively as BF16 without implementing a BF16 training run for comparison, which would completely negate the purpose of using FP8 for training in the first place. Even if FP8 training is practical for carefully selected hyperparameters under specific conditions, we assert that the costs of finding such conditions and the risks involved in training a less stable model outweigh the benefits of using FP8 for accelerated computation. Instead, we propose methods to evaluate the stability and robustness of reduced-precision training, which is vital for FP8 or other reduced-precision training schemes to be viable for real-world LLM training.

Also, we would like to preempt misunderstanding by clarifying that we are not questioning the usefulness of FP8 for inference. Several recent works (Lee et al., 2023; Kwon et al., 2022; Kim et al., 2023a) have shown that it is even possible to quantize LLM weights to below 4 bits without sacrificing much accuracy. Xia et al. (2024) has also shown that, even for the A100, using FP6 for inference is a viable option. We believe that dedicated FP8 and FP4 processors can make LLM inference simpler to implement and faster to compute.

From our experiments, several methods naturally suggest themselves as possible stabilization techniques. First, the initial stages of training could be conducted in higher precision, similar to how smaller batch sizes may be used during the initial stages of training as in Keskar et al. (2017). Increasing the precision when the loss landscape becomes too sharp may also provide a tradeoff between training speed and stability. Second, the more sensitive layers may be kept at high precision, while only the less sensitive layers are computed with reduced precision. For example, during our experiments, we found that removing masking from the LM head of a Llama model was sufficient to enable E7M7 training, although the resulting model was less stable. For GPT models, we found that increasing the precision of the first two decoder blocks to TF32 was sufficient to prevent loss divergence. However, as such compensatory techniques depend on the model architecture, training data, and other aspects of the training environment, we leave their investigation to future work.

## 6 LIMITATIONS

A limitation of this work is that it focuses on the initial stages of pre-training when many instabilities are known to arise only later in training (Bekman, 2023). For example, Wortsman et al. (2024) show that the logits of the outputs diverge from zero only at the later stages of training. To this, we argue that our studies likely underestimate the instabilities that FP8 or other reduced precision training schemes will face, further strengthening our case that reduced-precision training methods are too unstable to be profitably utilized in their current form.

Second, despite finding that the exponent bits are of greater importance to LLM training than mantissa bits, we were unable to experiment by increasing the number of exponent bits. This was because matrix multiplication in FP64 is over an order of magnitude slower than BF16 on A100 and H100 GPUs when using tensor cores. Experiments using representations such as E11M4, created by removing 48 mantissa bits from FP64, may be illuminating, but we found it impractical to train models with a greater number of exponent bits.

Finally, our experiments are limited in that they only the training loss is used as an evaluation metric instead of real-world natural language tasks such as MMLU (Hendrycks et al., 2021) scores. However, while lower perplexity is no guarantee of superior performance on downstream tasks, we believe that the divergence of the training loss is sufficient as an indicator of training failure.

## 7 CONCLUSION

We demonstrate that the training stability of LLMs decreases incrementally with the reduction of floating-point bit widths used for training models of up to 7B parameters. Using our proposed loss landscape sharpness metric, we measure the gradual increase of instability that leads to loss divergence, shedding light on a phenomenon with potentially large financial and environmental costs.

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

# A  APPENDIX

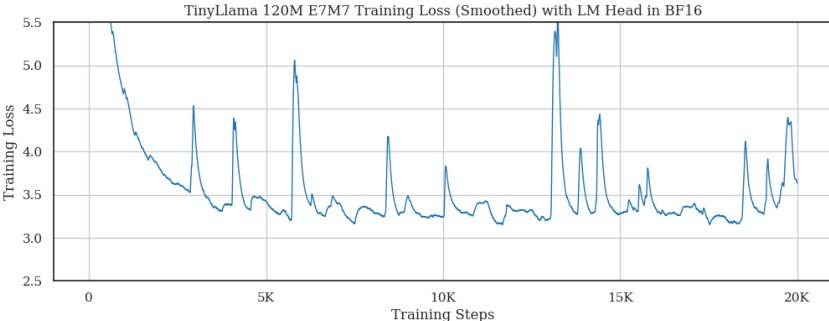

Figure 10: The training loss of a TinyLlama 120M model with clipped exponent at E7M7, excluding the LM head. The training loss is smoothed using exponential moving averages for better visualization. The results show that the exponent clipped models remain unstable when models in Figure 9 have stabilized after the same number of training steps.

Table 2: Robustness of the sharpness metric to $\epsilon$. We have found empirically that the loss landscape sharpness metric is robust to the choice of $\epsilon$. Below, we include a table with sharpness values for a wide range of $\epsilon$ values for a Llama 7B model trained for 5,000 steps. We use a different checkpoint from the one in Table 1 of the paper to demonstrate reproducibility.

| $\epsilon$ | Precision | 1K | 2K | 3K | 4K | 5K |
|---|---|---|---|---|---|---|
| 5.00E-05 | E8M3 | 0.02 | 0.06 | 0.18 | 0.19 | 0.17 |
| | E8M4 | 0.02 | 0.04 | 0.08 | 0.13 | 0.17 |
| | E8M5 | 0.02 | 0.03 | 0.04 | 0.05 | 0.07 |
| | E8M6 | 0.02 | 0.02 | 0.02 | 0.02 | 0.03 |
| | E8M7 | 0.02 | 0.02 | 0.02 | 0.02 | 0.02 |
| 1.00E-04 | E8M3 | 0.04 | 0.11 | 0.36 | 0.38 | 0.33 |
| | E8M4 | 0.04 | 0.08 | 0.16 | 0.26 | 0.34 |
| | E8M5 | 0.04 | 0.05 | 0.07 | 0.10 | 0.14 |
| | E8M6 | 0.04 | 0.04 | 0.04 | 0.05 | 0.05 |
| | E8M7 | 0.03 | 0.04 | 0.04 | 0.04 | 0.04 |
| 5.00E-04 | E8M3 | 0.19 | 0.51 | 1.70 | 1.80 | 1.49 |
| | E8M4 | 0.18 | 0.37 | 0.74 | 1.22 | 1.54 |
| | E8M5 | 0.18 | 0.25 | 0.34 | 0.48 | 0.64 |
| | E8M6 | 0.18 | 0.21 | 0.21 | 0.23 | 0.25 |
| | E8M7 | 0.16 | 0.18 | 0.17 | 0.19 | 0.19 |
| 1.00E-03 | E8M3 | 0.38 | 0.98 | 3.31 | 3.59 | 2.81 |
| | E8M4 | 0.36 | 0.71 | 1.42 | 2.32 | 2.86 |
| | E8M5 | 0.34 | 0.49 | 0.66 | 0.92 | 1.21 |
| | E8M6 | 0.34 | 0.41 | 0.40 | 0.44 | 0.48 |
| | E8M7 | 0.30 | 0.35 | 0.34 | 0.37 | 0.38 |
| 5.00E-03 | E8M3 | 1.73 | 4.47 | 13.64 | 11.64 | 9.58 |
| | E8M4 | 1.61 | 3.26 | 6.43 | 9.87 | 11.39 |
| | E8M5 | 1.55 | 2.25 | 2.92 | 4.10 | 5.28 |
| | E8M6 | 1.55 | 1.87 | 1.85 | 2.03 | 2.21 |
| | E8M7 | 1.36 | 1.60 | 1.53 | 1.71 | 1.73 |

Table 3: We show the loss landscape sharpness of a Llama 7B model initially trained with E8M3 precision for 6,000 training steps that was then trained with standard BF16. We can see that the sharpness indicator decreases in value with more training on BF16, indicating that it is capturing the increased stability of training that comes with BF16 over E8M3.

| Train Step | Sharpness |
|---|---|
| 7K | 1.35 |
| 8K | 1.14 |
| 9K | 0.98 |
| 10K | 0.90 |
| 11K | 0.87 |
| 12K | 0.77 |
| 13K | 0.71 |
| 14K | 0.63 |
| 15K | 0.60 |
| 16K | 0.59 |
| 17K | 0.57 |
| 18K | 0.50 |
| 19K | 0.48 |
| 20K | 0.48 |
| 21K | 0.46 |
| 22K | 0.44 |
| 23K | 0.40 |
| 24K | 0.40 |
| 25K | 0.37 |
| 26K | 0.34 |
| 27K | 0.34 |
| 28K | 0.34 |
| 29K | 0.33 |
| 30K | 0.35 |
| 31K | 0.32 |
| 32K | 0.32 |
| 33K | 0.30 |
| 34K | 0.29 |
| 35K | 0.29 |
| 36K | 0.28 |
| 37K | 0.27 |
| 38K | 0.26 |
| 39K | 0.27 |
| 40K | 0.27 |
| 41K | 0.25 |
| 42K | 0.23 |
| 43K | 0.23 |
| 44K | 0.24 |

```python
def backward(inputs, weight, output_gradient):
    masked_inputs = reduce_precision(inputs)
    masked_weight = reduce_precision(weight)
    masked_output_gradient = reduce_precision(output_gradient)
    inputs_gradient = F.linear(masked_inputs, masked_weight.T)
    weight_gradient = F.linear(masked_output_gradient.T, masked_weight.T)
    masked_inputs_gradient = reduce_precision(inputs_gradient)
    masked_weight_gradient = reduce_precision(weight_gradient)
    return masked_inputs_gradient, masked_weight_gradient
```

Figure 11: PyTorch-like pseudocode for the reduced-precision backward pass.

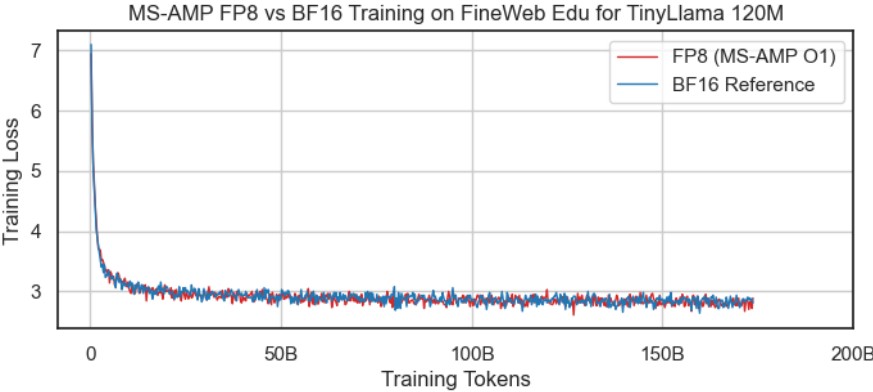

Figure 12: A comparison between MS-AMP FP8 training (O1) and BF16 training on a subsample of the FineWeb Edu (Penedo et al., 2024) dataset. We find that when the model, Llama 120M for this experiment, is trained on a "clean" dataset such as FineWeb Edu, the divergence between MS-AMP FP8 O1 and BF16 disappears. However, LLM pretraining datasets in production environments are usually much "dirtier" than those of popular open-source datasets. For example, extensive data filtering may not be an option for low-resource languages. Also, for newer domains such as video or robotic motion, well-established metrics of data quality do not yet exist. Therefore, the finding that FP8 training works well on "clean" data supports our claim that hidden instabilities exist in reduced-precision training rather than disproving it.

