# OpenReview forum: "To FP8 and Back Again: Quantifying Reduced Precision Effects on LLM Training Stability"
_ICLR.cc/2025/Conference — Submitted to ICLR 2025_

### Official Review · Reviewer_nxKU · 2024-10-27

**Soundness:** 2
**Presentation:** 2
**Contribution:** 2
**Rating:** 3
**Confidence:** 4

**Summary:**

The paper investigates training using low precision, specifically the FP8 datatype. It examines the stability of FP8 training across various seeds and initializations, revealing loss spikes during certain training phases. To further understand these issues, the authors analyze training stability by simulating different numbers of mantissa bits. Finally, they propose a method to assess loss sharpness and predict potential training divergence.

**Strengths:**

The paper suggest a sharpness metric to help to check where a model training will diverge  The the authors claims that the proposed method is more suitable for autoregressive models. I think that comparison to previous method, showing cases where the proposed method works better than previous one in autoregressive models is a must.

**Weaknesses:**

I think the weaker part of the paper is the experiment section, mainly  the main MS-AMP experiment (section 4.1). - which is the main motivations for all the paper.
The authors claims that in all cases FP8 training is not able to get the same results as the baseline (BF16) counterpart. We can find 2 papers - from Microsoft (https://arxiv.org/pdf/2310.18313) and Intel (https://arxiv.org/pdf/2409.12517) showing that is is possible to train LLM in FP8 - I suggest the authors to investigate the gap between their paper and these.

**Questions:**

1) Please add comparison to previous sharpness metric, showing why the proposed method is more suitable for autoregressive models.
2) Does the bit reduction experiments include scaling when reducing the exponent bits. If not, I suggest to add it- I believe in that cases we will see completely different conclusions.
3) How do you implement the experiment of reducing mantissa bits? For the exponent it is well explained in the paper (clipping). Moreover, the author claim their method of masking is an imperfect approximation - can you please explain what is the gap? How do you show this gap is not critical.

4) Please try to clarify why the authors are not able to converge FP8 training while 2 previous papers are able to converge.


===================================================================

Rebuttal:

Dear Authors,

Thank you for taking the time to address the questions and concerns raised.

Unfortunately, I believe the primary issue remains unresolved. The authors attribute the divergence in their FP8 training to the use of an open-source dataset. However, Intel’s paper demonstrates successful FP8 convergence using open-source datasets as well. Additionally, while the authors suggest that Intel’s findings support their claims, I believe this is not accurate. In Intel’s work, the divergence is attributed to the use of the SwiGLU activation function, which is not employed in this paper (as nanoGPT uses the GeLU activation function).

Furthermore, I believe that the absence of scaling and the bias introduced by the bitmask in the mantissa (which could be critical during the backward pass) may significantly impact the conclusions drawn in this paper.

I will maintain my score.

---

> ### Author Response · Authors · 2024-11-21
> **Reply to Feedback by Reviewer nxKU.**
>
> Thank you for carefully critiquing our work and methodology. We address the weaknesses mentioned and hope our response is helpful.
>
> **Weaknesses**:
>
> We thank the reviewer for the survey of FP8 training. Indeed, the two papers mentioned are most relevant to this work. The first paper from Microsoft proposed the MS-AMP method, and we analyze the results of applying MS-AMP to the nanoGPT codebase in Figure 6 of our work. MS-AMP O1 in our work is equivalent to the FP8 #2b method in the MS-AMP paper.
>
> The second paper from Intel was published more recently, and we were thus unable to apply their Smooth-SwiGLU and FP8 optimizer state conversion methods.
>
> The work by Intel, however, supports our claim that FP8 training is not equivalent to BF16 training rather than dispelling the concerns that we raise. Figure 2 of their work shows that although FP8 initially has no conversion issues compared to BF16, a divergence emerges after 200B tokens of training for Llama 7B models. Although we could not conduct such extensive experiments due to a lack of resources, this is precisely the type of issue that our work aimed to raise awareness of. The authors propose adding scaling factors to the SwiGLU activation to prevent divergence.
>
> Our work aims to shift the focus of FP8 training research away from whether FP8 training can be made to work from an engineering perspective and towards discussions of whether FP8 is worth using from a financial perspective. The main advantage of using FP8 is that it reduces training costs by improving throughput. If the quality of the model deteriorates or requires additional training to compensate, the financial incentives to deploy FP8 are cast into doubt. We do not ask, “Can it be done?” but instead, “At what cost?”.
>
> Production environments often cannot use thoroughly cleaned and filtered datasets for which the optimal learning rates have already been discovered for given model sizes. For new types of data, such as video, standards for how data is to be curated are yet to be established as rigorously as for text data. Even for text data, training on low-resource languages or specialized fields may require using all available data, which can significantly increase training instability. We seek to know how new methods will perform on rough, untrodden paths, not only on well-paved roads.
>
> **Questions**:
>
> 1. Thank you for raising this point. The value of the sharpness indicator as an indicator of training instability is indeed an important issue. In Figure 2 of our work, we show the results for visualizing loss landscape sharpness using the method proposed by Li et al. We found no signs of sharpness when using this method, even when the model was undergoing divergence. As discussed in the reply to all reviewers, we sought additional verification of our proposed indicator by measuring the loss landscape sharpness when a model (Llama 7B) was initially trained with reduced precision (E8M3) but later trained with BF16. We found that the sharpness decreased, strengthening our case that it is a useful indicator of training instability.
>
> 2. For most of our experiments, we only clip the mantissa bits and do not add scaling factors. For the experiment in Figure 10 of the appendix, we removed an exponent bit and found that the effect was much greater than when we removed a mantissa bit. As FP16 with scaling is possible, we agree that adding scaling to the exponent clipping would change the result. However, the experiments with intermediate bits aimed to see the effects of increasing instability in higher resolution. We wished to see incrementally increasing instability effects at intermediate bits instead of simply comparing 16-bit and 8-bit schemes. Analyzing reduced exponent bits with scaling factors would be a valuable addition to this work, and we hope to explore this direction in the future.
>
> 3. We apply a bitmask to the mantissa to remove the least significant bits, equivalent to rounding down the mantissa. While this may introduce a slight bias, the effect is smaller than the rounding error introduced for FP8 types, which have 3-bit and 2-bit mantissa widths for E4M3 and E5M2, respectively.
>
> 4. Thank you for this important question. In exploring the answer, we discovered that data quality was the likely cause of the divergence shown in Figure 6 between baseline BF16 and MS-AMP FP8 training. As noted in the reply to all reviewers, when the FineWeb Edu dataset, a clean dataset with high educational content, was used for training, the differences in training loss disappeared. As discussed in the weaknesses section, we consider this a validation of our argument that FP8 is not equivalent to BF16. The Intel results show divergence even for standard FP8 training only after training on 200B tokens on a Llama 7B model, which is beyond our computational resources. Also, the proposed solution involves architectural modifications to reduce the effects caused by correlations in the SwiGLU weights.

---

> > ### Author Response · Authors · 2024-12-04
> > **Further Reply to Feedback**
> >
> > Thank you for your review of our replies. We would like to answer the further points that were raised.
> >
> > 1. We believe that the divergence in training loss appeared in the nanoGPT MS-AMP results because of the data quality, not because the dataset was open-source per se. The FineWeb Edu dataset is also open source but widely acknowledged to be of high quality.
> >
> > 2. We believe that the results from FP8 training from Intel support our work because it shows in Figure 2a that some instabilities only emerge much later in training. This is in line with our claim that FP8 causes hidden instabilities that must be further investigated to show equivalence. Although newly proposed methods will surely improve said stability, we hope to investigate, or perhaps inspire others to investigate, whether these methods are as robust as the current best practice. Even if differences only emerge in edge cases, we believe that identifying such rough edges is a priority for the practical deployment of FP8 in LLM training.
> >
> > 3. Because clipping the mantissa does not change the range of expressible values, only the resolution of values, the absence of a scaling factor does not affect the results. For example, BF16 has the same number of exponent bits as FP32, meaning it can be used instead of FP32 without a scaling factor, though the resulting values are coarser. We use the scaling factors decided by the MS-AMP library for our MS-AMP experiments.
> >
> > Best wishes,
> >
> > The authors.

---

### Official Review · Reviewer_mgM3 · 2024-10-29

**Soundness:** 2
**Presentation:** 2
**Contribution:** 2
**Rating:** 5
**Confidence:** 3

**Summary:**

This paper explores the feasibility of using reduced-precision floating-point representations, specifically FP8, for large language model (LLM) training to enhance computational efficiency. While BF16 has become standard due to its balance of precision and performance, FP8 poses challenges in stability. The authors propose a new metric to quantify loss landscape sharpness, aiding in predicting training stability. They analyze the effects of reducing mantissa and exponent bits, finding that exponent bit reduction significantly impacts model performance. The contributions include a novel approach to evaluating training stability and insights into the trade-offs of using lower precision in LLM training.

**Strengths:**

-The motivation of this paper is compelling, particularly the concern regarding the stability of FP8 compared to FP16 and BF16. This raises important questions about the cost-effectiveness of FP8 for LLM training. A deep analysis and understanding of the effects of low-bit training can significantly benefit the research community by providing valuable insights.
-The introduction of a new metric for quantifying loss landscape sharpness is good. This analytical framework could be beneficial for the research community.

**Weaknesses:**

My primary concern with this paper is the confidence in the experimental validation, especially given the smaller scale of the models used. Theoretically, we understand that reduced precision can affect model convergence. However, my intuition suggests that as models grow larger and datasets become more extensive and of higher quality, the models' robustness improves, potentially mitigating the impact of low-bit precision. This phenomenon is also observed in low-bit post-training quantization, where larger models are less affected by quantization. I am uncertain if the observed effects would persist in larger models.

As this paper is based on MS-AMP. The MS-AMP paper, which tested with 7B, 13B, and 175B models on more diverse datasets, concluded that FP8 training loss is consistent with FP16. This seems contradictory to the findings of this paper. Is there a difference in experimental settings, or did I miss something?

**Questions:**

-Many previous works have proposed techniques to address numerical instability, such as precision decoupling and automatic scaling in MS-AMP. Were these techniques or similar techniques applied in your experiments, particularly in the simulations conducted in sections 4.2 and 4.3?

-I'm unsure about the data quantity and quality used in this work, as the experimental settings aren't clearly described. For instance, it's unclear what dataset was used in Figure 6, and the training steps don't intuitively reflect how many tokens were trained.

---

> ### Author Response · Authors · 2024-11-21
> **Reply to Feedback by Reviewer mgM3.**
>
> Thank you for your constructive critique of our work. Due to the feedback, we could conceive new experiments to explore aspects of our work in greater depth. We hope to address the raised issues in the following response.
>
> **Weaknesses**:
>
> The two areas for improvement mentioned in the review were (1) concerns about whether the reduced-precision issues would persist in larger models and (2) why the results in this work diverged from those from the MS-AMP paper.
>
> For the first point, our experiments using Llama 7B models suggest that sensitivities to lower precisions persist even in larger models. As we can see even Llama 7B models show increased training instability when using reduced precision, it is not implausible that the issue persists even in larger models. In addition, although the largest LLMs may reach hundreds of billions or even trillions of parameters, there has been a trend to utilize smaller models of only a few billion parameters on larger datasets, especially for limited tasks. We believe that our work may also inform researchers conducting training work in such areas.
>
> For the second point, we found that the critical issue was the data quality involved in training. In our experiments, we used the OpenWebText dataset from the nanoGPT repository to reproduce the results for MS-AMP, the results of which are shown in Figure 6 of our work. However, Section A.3 of the Appendix in the MS-AMP paper notes that the pretraining data used for experiments consisted of a large custom dataset with extensive filtering and curation, which was not made publicly available.
>
> As mentioned in the comment to all reviewers, we suspected that data quality might be the cause of the divergence and attempted training on a subsample of the FineWeb Edu dataset, which was also extensively filtered and curated for educational content by training on smaller models. We found that the difference between BF16 training and MS-AMP training disappeared after changing the dataset, indicating that the difference in convergence was caused by the lower quality of the OpenWebText dataset.
>
> While this finding may appear to weaken our argument, we argue instead that it strengthens it. In actual training environments, we cannot simply take extensively verified open-source datasets and train them on models for which the optimal learning rate is already known. Low-resource languages may require using any available data, while it is unclear how new domains, such as robotic motion, will interact with the other data during training. We do not argue that FP8 training of LLMs is impossible. However, we must consider that the primary motivation for using FP8 over BF16 is to reduce costs. If costs arising from other sources outweigh the costs saved from the increased throughput, there is no justification for using FP8 over BF16. Considering that the speed boost obtained by applying FP8 to training is, at best, approximately 50% faster, we believe that works studying the use of FP8 for LLM pretraining should address how their methods perform when stepping out of well-trodden paths. If nothing else, we hope to move the conversation from discussing how FP8 training methods could be made to perform well on some well-known tasks to focusing on cost-effectiveness.
>
> **Questions**:
>
> 1. We conduct our experiments on two fronts. First, in Figure 6 of our work, we show the results of applying MS-AMP to nanoGPT training. However, because we wished to gain a more fine-grained understanding of the destabilizing effects of precision reduction, we conducted our other experiments using clipped mantissa values on Llama models of various sizes.
> 2. Thank you for pointing out this crucial issue. As mentioned above, this question has helped us examine a new axis of data quality in our work. To answer the question, we used the datasets in the respective repositories. For the experiments with nanoGPT, we use OpenWebText, an open-source reproduction of the GPT-2 dataset. For the TinyLlama experiments, we use a mix of the SlimPajama and Starcoder datasets in a ratio of approximately 7:3. The number of tokens per training step can be calculated using the equation $training\ step \times sequence\ length\times global\ batch\ size$.

---

> > ### Comment · Reviewer_mgM3 · 2024-11-27
> > **My primary concern not addressed**
> >
> > Thank you for the detailed feedback. However, I find that your response does not directly address my primary concern. Instead, it seems to align more closely with my original intuition: **"as models grow larger and datasets become more extensive and of higher quality, the robustness of models improves, mitigating the impact of low-bit precision"**.
> >
> > I do not see sufficient evidence or rationale to support the claim: **"it is not implausible that the issue persists even in larger models."** On the contrary, existing evidence suggests that this issue is mitigated in larger models. If you have evidence to support your position, please share it.
> >
> > In terms of model size, I would expect a comparison across models of different sizes to see whether larger model has better or worse training stability, of course using the same (high-quality) dataset. (I understand the training cost, you can consider 1B,3B,7B size. And even with very limited training tokens, i guess we can see some trend.)
> >
> > In terms of data quality, in pretraining, data engineering is a critical step, so that no one would begin training a large-scale model without thoroughly cleaning and curating the dataset. This is a standard practice in the field. It weakens the argument to focus on suboptimal practices with lower quality.

---

> > > ### Author Response · Authors · 2024-11-27
> > > **Further Reply to Feedback**
> > >
> > > Thank you for clarifying the issues that have raised your concern. We hope to address these issues as best we can.
> > >
> > > First, to answer the question concerning larger models, our experiments with mantissa clipping use Llama 7B models in experiments shown in Figure 8, Table 1, and Table 3 of the appendix. In these experiments, we find similar results with Llama 120M models that use the same dataset, finding that by 5K training steps, E8M3 had already diverged, and E8M4 was beginning to diverge. However, we could only train Llama 7B models for 5K training steps for all configurations due to resource constraints. In Table 3, we show loss landscape sharpness results for Llama 7B, initially trained with E8M3 for 6K training steps, then trained with BF16 for the rest of the training. From these experiments, we may conclude that our findings concerning the effects of mantissa reduction hold, even in Llama 7B models. Also, because these models all use the SlimPajama and Starcoder dataset mix used by TinyLlama, we believe that data quality was not too much of an issue.
> > >
> > > Second, we only performed experiments using small nanoGPT models for MS-AMP for two reasons. (1) Our aim is not to “disprove” any specific method but to show that instabilities, which might not be initially visible, are greater than what the cost savings may justify. We therefore only investigated MS-AMP using small models as we immediately found a difference. (2) The TinyLlama implementation uses PyTorch Lightning with FSDP via the Lightning Fabric API, making the application of MS-AMP to larger models such as Llama 7B challenging. Such models cannot be trained simply using DDP, and applying MS-AMP via FSDP could not be handled by the Lightning Fabric wrapper.
> > >
> > > In summary, our previous discussion concerning MS-AMP mainly sought to find why we could not reproduce the results in [1], which we now believe is due to the low quality of the OpenWebText dataset used in the nanoGPT repository. For our mantissa clipping experiments, where we investigate the increase in instability in greater detail, we verify our results on models up to Llama 7B, if only for very early training stages.
> > >
> > > Third, concerning the issue of data quality, we would like to refer to the results in Figure 2 (a) of [2]. Despite being trained on the same RedPajama dataset, the initial experiment using FP8 only begins to diverge at 200B tokens. Moreover, this fact is only apparent because we have a BF16 model trained side-by-side with the FP8 model. The loss divergence would be clear by 300B tokens, even without the BF16 training run for comparison. However, a considerable amount of resources would have been expended by this time. If FP8 training were to be deployed in real-world training, there would be no BF16 training run for comparison, making it much more costly to identify the issue.
> > >
> > > Finally, we would like to restate our case that we aim to show that more work is required to show that FP8 training is **equivalent** to BF16. We do not argue that FP8 training is always inferior to it. Even if the effects of FP8 were found only to emerge in “low-quality” data, this would be a significant finding showing that FP8 training has a narrower hyper-parameter space. This means that models that use multilingual or multimodal data for training would have to check whether their FP8 training produced results similar to those of their BF16 runs. This is an important consideration from a cost perspective, as many would choose not to take the risk. Because the purpose of FP8 is to reduce costs, not boost model quality, such considerations must be taken into account. We do not aim to show that FP8 training “does not work”, we aim to show that FP8 works “under narrower circumstances” than BF16. Furthermore, we hope to motivate future research in this field to account for such issues, giving greater confidence to users who must risk many millions of dollars on training going well.
> > >
> > > [1] FP8-LM: Training FP8 Large Language Models
> > >
> > > [2] Scaling FP8 training to trillion-token LLMs

---

### Official Review · Reviewer_9xNk · 2024-11-03

**Soundness:** 2
**Presentation:** 2
**Contribution:** 3
**Rating:** 6
**Confidence:** 2

**Summary:**

The paper proposes a new metric to evaluate the training instabilities of LLMs when reducing the floating precision of the model's weights. The paper introduces a novel loss landscape metric based on the previously proposed Keskar et al. (2017). However, the paper's novel metric is computed on the last token's logits rather than on the model's input for the computation. Thanks to this change, the computational cost of computing the metric is significantly reduced since the metric has to be computed only once for each measurement. This choice is motivated by the autoregressive decoder-only model's last token, which receives inputs from all other tokens.

By proposing this novel metric, the paper investigates the cost-effectiveness and impact of reducing the model's floating point precision. The paper analyzes the impact of reducing the model's weight precision regarding training instabilities by conducting an ablation study. They conducted experiments by simulating incremental bit reduction of the mantissa or exponent bits of the model's weights.

**Strengths:**

The proposed metric is an effective, computationally cheap, proxy measure of the underlying training instability. The paper clearly shows in Figure 8 and Table 1 that the proposed metric can detect the training divergence before it occurs for the E8M5 and compare the results with other reduced floating-point precision (for example)

The paper's outcomes confirm previous findings that neural network training is more sensitive to exponent bit reductions than to mantissa bits.

**Weaknesses:**

The paper proposes a novel and computationally cheap metric to detect training divergence based on the work of Keskar et al. (2017). However, the paper does not compare the performances or computational costs of previously proposed methods in the literature. Also, the paper does not consider or compare methods exploiting the gradient or the internal activations of the model to detect training divergences.

The paper does not consider different training hyperparameters from those reported at the end of the first paragraph of the "Method" section.

**Questions:**

- Could you maybe compare your proposed metric to the metric proposed in Keskar et a. (2017) or other SOTA metrics in terms of computational costs and early detection of training instabilities?

- Could you maybe run the experiments on more training schedules (i.e., initial learning rate, learning rate) than the ones mentioned in the first paragraph of the Methods section to verify the consistency of your observed results?

- Could you maybe include the results (i.e., a table or plot) showing the failure of training with lower exponent bits (e.g. E7E7)?

- Could you maybe verify the two suggestions (i.e., high-precision initial training stages and higher precision for sensitive layers and vice versa for less sensitive layers) proposed in the last paragraph of the "Discussion" section as possible training stabilization techniques?

---

> ### Author Response · Authors · 2024-11-21
> **Reply to Feedback by Reviewer 9xNk.**
>
> Thank you for your thoughtful review of our work. Below, we aim to address the points that have been mentioned.
>
> **Weaknesses**:
>
> The two points mentioned as weaknesses of our work are (1) that it needs to be compared with alternative methods of measuring training divergence and (2) that hyper-parameters other than the learning rate and model size should also be searched.
>
> 1. Regarding the first point, we initially attempted to use the loss landscape sharpness visualization technique proposed by Li et al. However, as shown in Figure 2 of our work, we found that the method was unsuitable for our work as the visualized loss landscape surface continued to be smooth even when the model was in the process of divergence. While we are unaware of other methods that measure training instabilities in autoregressive models, the experiment in our response to all reviewers supports our claim that the sharpness indicator proposed in this work can also detect decreases in training instability when a higher precision is used.
>
> 2. Regarding the second point, we explored the effect of using higher-quality data in our additional experiments. Although the training losses for MS-AMP FP8 and BF16 diverge for the OpenWebText dataset used for training nanoGPT, using a higher-quality dataset such as FineWeb Edu hides the difference. This finding supports our argument that FP8 introduces subtle instabilities in LLM training, weakening the financial case for deploying it.
>
> **Questions**:
>
> 1. Compared to Keskar et al., our method uses at least millions, possibly billions, of times fewer resources. The reasons are twofold. First, because we use only the last token of the autoregressive prediction, our method skips computation on the rest of the sequence. Second, because we apply noise to the output logits instead of the model’s input embeddings, we need not perform a model forward pass for each iteration of the L-BFGS-B algorithm, requiring only a pass through the cross-entropy loss layer.
>
> 2. Regarding this point, we focused on the learning rate as the main hyper-parameter because the learning rate is known to be a key hyper-parameter. Model size was also tested, but to a lesser extent, because increased model size requires greater resources. In addition to the learning rate, we attempt additional experiments on different datasets for this review. In these, we train MS-AMP models on a sample of the FineWeb Edu dataset, which has been extensively cleaned and filtered for “educational content”. We believe that these experiments show that data quality can also expose the reduced stability of FP8.
>
> 3. We include results for our naïve implementation of exponent clipping in Figure 10 of the Appendix.
>
> 4. We thank the reviewer for reading our work in depth. Further investigation in this area would be of interest. Some of our preliminary results are positive. For example, excluding the first two transformer blocks of nanoGPT models from low precision prevents them from collapsing. However, this trick did not affect Llama models. Also, although we did not try experiments where the model was trained in high precision first and later used low precision, we did attempt to train the model first in low precision, then revert to high precision. The results are shown in the table in our response to all reviewers.

---

> > ### Comment · Reviewer_9xNk · 2024-11-22
> > **Reviews Update**
> >
> > Dear Authors,
> >
> > Thank you for taking care to answer my questions and doubts. However, my comparison doubts still need to be improved, especially regarding the comparison to the SOTA FP8 training mentioned by other reviewers. Also, because the new results that verify the hypothesis that: "data quality was the likely cause of the divergence" might need more experiments and analysis, I will keep the Rating Score unchanged.
> >
> > Best regards.

---

> > > ### Author Response · Authors · 2024-11-23
> > > **Comment to Update**
> > >
> > > Thank you for your prompt response. We are heartened that some doubts have been addressed.
> > > Realizing that data quality is a key factor was a crucial insight. We appreciate all the reviewers for highlighting this important element of our research.
> > >
> > > Regarding the comparison to SOTA FP8 training methods, those that we are aware of are (1) Transformer Engine, (2) MS-AMP, and (3) the work by Intel. FP8 training with Transformer Engine produced poor results, and we did not include them in our analysis. The work by Intel was published in conjunction with ours, and open-source implementations with plugins for experimentation are not yet available.
> > >
> > > We wish to emphasize again that our work does not involve showing that FP8 training “does not work”. On the contrary, we are quite confident that the newly proposed FP8 training methods work well **within the bounds the authors assumed**. We argue that because the primary motivation behind FP8 training is cost reduction, not an improvement in the model quality, those proposing such methods should “stress test” them to ensure that their new methods work well not only on well-trodden paths but also on the rugged environments that real-world model training will occur in.
> > >
> > > Our analysis of greater instability for higher learning rates reflects the concern that the learning rate may not be optimally tuned for a particular task for a specific model architecture and size. We propose methods such as the loss landscape sharpness indicator as a possible tool for investigating hidden instabilities without incurring the costs of training models until they diverge. Hopefully, future work into low-precision training will incorporate such analyses to produce more robust results that users can use with confidence.

---

### Official Review · Reviewer_Nnvy · 2024-11-04

**Soundness:** 2
**Presentation:** 2
**Contribution:** 2
**Rating:** 5
**Confidence:** 2

**Summary:**

The authors explores what happens when large language models (LLMs) are trained using FP8 precision, meaning only 8 bits are used for floating-point numbers. FP8 is theoretically cheaper, but it introduces significant instability compared to BF16, the precision currently preferred for these models. The authors introduce a new metric to measure training stability by examining "loss landscape sharpness," which helps predict when training might diverge. They found that FP8 limits the range of hyperparameters that work, reducing its potential cost-savings. Overall, FP8 isn’t quite suitable yet for full-scale LLM training without further stabilization methods.

**Strengths:**

1. It’s got this new loss landscape sharpness metric that basically acts like an early-warning system for training issues, spotting instability before it even shows up in the loss curve which helps predict when training might go off-track.

2. Unlike some of the latest studies that stick to small-scale models, this paper actually dives into FP8 precision with full-size LLMs, so the findings are more practical and closer to real-world training needs.

3. They tried out reducing bits both on exponent and mantissa, making it unique since other papers usually just mess with mantissa bits, giving a fuller picture of how bit reduction affects training stability.

4. The paper shows in real-world training that FP8 doesn’t handle hyperparameters as well as BF16, which limits its ability to actually save money in training, something that hasn’t been looked into much in other studies.

5. By using TinyLlama for their experiments, this study connects the theory of low-bit precision with real stability checks, giving a strong framework for researchers wanting to make low-precision training more stable in LLMs.

**Weaknesses:**

- The study mostly measures training stability by focusing on loss landscape sharpness and tracking divergence in training loss, which is useful but doesn’t fully reveal how well the model would perform on real-world tasks like NLP benchmarks. This narrow focus on training metrics rather than practical testing limits the broader application of the findings in real scenarios.

- They mention computational limits for testing with exponent bits, so they mainly tried reducing bits without exploring the impact of increasing them. Some recent research suggests adding exponent bits could help stabilize FP8, leaving an area of potential solutions unexplored.

- Because running full-scale experiments is costly, they used smaller models like TinyLlama to approximate larger ones. But recent studies point out that these smaller models don’t always behave the same as big ones like GPT-3, so the results might not directly translate to large-scale models.

**Questions:**

- In Figure 6, where MS-AMP was applied to nanoGPT with and without the LM head in FP8, what does the observed performance degradation indicate about the LM head’s role in stability? And how do the loss trends compare to BF16 training based on the exponential moving averages?

- Table 1 shows loss landscape sharpness values for Llama v2 7B models at different precisions (E8M3 to E8M7) over various training steps; how do these sharpness trends reveal the model's progress towards loss divergence, and are there any specific sharpness thresholds that clearly predict instability?

- The stability of E8M5 configurations is tested under different learning rates in Figure 9, showing more frequent loss spikes at higher rates—how does this pattern of sharpness changes affect hyperparameter tuning in low-precision training, and what insights do these learning rate sensitivities give for FP8 compared to BF16?

---

> ### Author Response · Authors · 2024-11-21
> **Reply to Feedback by Reviewer Nnvy**
>
> Thank you for your thoughtful feedback on our work. We hope that our reply addresses the points raised by the reviewer.
>
> **Weaknesses**:
>
> 1. As noted in our limitations section, we wholeheartedly agree that training loss alone is insufficient to determine a model’s quality. Unfortunately, because we primarily analyze our models in the early stages of training, making meaningful comparisons on real-world benchmarks is challenging. Models attain meaningful scores only when they have been trained more extensively. This will be an area of further research. However, a model with a divergent loss function will not obtain a good score on downstream benchmarks, making good training loss conversion a necessary, if not sufficient, condition for LLM quality.
>
> 2. Exploring reduced exponent bits with common scaling factors would be a significant addition to our research. This would involve the analysis of algorithms to decide the granularity and the values of the scaling factors, adding another dimension of exploration. However, for simplicity of analysis, our experiments mainly analyze the effects of reducing mantissa bits from BF16, sidestepping issues that may arise from the reduced exponent range compared to FP32. Because of this, scaling problems do not affect our experiments on mantissa clipping. We hope that our experiments with MS-AMP, which implements scaling and other techniques, will be informative on this front.
>
> 3. Due to resource considerations, we were limited to models of 7B parameters or lower. However, recent trends in LLM training have enabled even relatively smaller models, such as Llama v3.1 8B, to greatly improve in capabilities. The latest generation of Llama v3 models have increased the number of training tokens applied to Llama 8B by orders of magnitude, finding that scaling laws still hold. Because of high inference costs, much research has gone into improving the performance of smaller models. Many applications, especially in restricted domains, are well served by these smaller models. As such, our work can benefit real-world use cases as well as serve as the basis for work on larger models.
>
> 4. Finally, we note the absence of FP16 in recent LLM training. Although early models such as OPT used FP16 with scaling factors for model training, nearly all LLMs made public within the last two years have used mixed-precision BF16 for training despite BF16 having three fewer mantissa bits and thus providing lower resolution. Though this only constitutes circumstantial evidence, the transition from FP16 to BF16, despite the former’s superior resolution, strongly suggests that large models are still prone to instabilities, even when scaling is applied. When considering that both E4M3 and E5M2 have less expressive capacity than FP16, we believe that a rigorous investigation of training robustness is due.
>
> **Questions**:
>
> 1. The considerable divergence in loss curves for FP8 training that arises depending on whether the LM head is included or excluded indicates the importance of even small errors at the output logits. This is in line with previous work on post-training quantization, which has found that the embedding layer and the final projection layer, which are sometimes tied together, are highly sensitive to quantization.
>
> 2. In Figure 6 of our work, we compare applying MS-AMP to the model excluding the LM head in red and show the baseline BF16 results in blue. As indicated by the black horizontal line, the training loss of the model trained using MS-AMP does not converge to the same level, even after 6 times the number of training steps. This indicates, at least for the conditions examined in Figure 6, that FP8 is less capable of training convergence than BF16. In additional experiments conducted for the review, we find that when a “cleaner” dataset, namely the FineWeb Edu dataset, is used, this divergence disappears, strengthening our claim that FP8 narrows the hyper-parameter space where stable training can occur.
>
> 3. Although we were unable to find specific threshold values where model collapse suddenly occurs, we were able to confirm that the sharpness is a valid indicator of training instability. In the reply to all reviewers, we find that the loss landscape sharpness decreases when we train a model that has already been trained in low precision in the original BF16. We summarize the results in the attached table.
>
> 4. The experiments for E8M5 help demonstrate one of the conditions that can reveal the hidden instability of reduced-precision training, namely, increased sensitivity to learning rate changes. Although it may be possible to find the optimal learning rate for each model and dataset, we argue that reduced-precision methods, even if they work well under best-case scenarios, are more sensitive to the choice of hyper-parameters, which can be very expensive to find for LLMs.

---

### Author Response · Authors · 2024-11-20
**Reply to Reviews from All Reviewers.**

We sincerely appreciate the feedback and comments from all reviewers on our work. The feedback has inspired us to conduct new experiments and measurements, which have led to several new findings. In the following discussions, we aim to address several weaknesses in our work.

First, concerns were raised that we could not reproduce the results in the MS-AMP paper, which shows equivalent results for GPT-2 124M training for MS-AMP O1 and BF16. The only significant difference between the experimental settings in the MS-AMP paper and our work was the dataset, which we cannot access. This raised our suspicion that the quality of data might be a factor in revealing hidden instabilities. FP8 might produce equivalent results on “clean” datasets with well-tested hyper-parameters but diverge on “dirty” data with more challenging distributions.

To verify our hypothesis, we used a sample of the FineWeb Edu dataset, which has been extensively filtered for “educational” content and verified with training runs on smaller models. The training loss curves for MS-AMP and BF16 converged with no noticeable difference. Figure 12 of the Appendix shows our results for this experiment.

This result validates rather than invalidates the main thrust of this work. In practice, pretraining data often cannot undergo the same level of filtering as open-source datasets such as FineWeb. Also, introducing different types of data, such as multilingual or multimodal inputs, may introduce differences in the distribution that require additional stabilization.

Because the primary purpose of deploying FP8 in training is to save costs by reducing training time, not improving the end performance of the model, a change in the conversation is needed. Our claim in this work is not that FP8 training in LLMs is impossible but that it is not equivalent to BF16 training when considering the complexities of real-world training. In practice, other works find an increase of 30~50% in training throughput when applying FP8 training. However, an investigation of these methods is needed to verify whether they hold up in actual training, where the learning rate is likely not to be tuned optimally to the target model and dataset while the inputs contain data from various sources and are of varying quality. If these cause an increase in training instability or require additional training steps to converge, the financial incentives to apply FP8 for LLM training become much weaker.

Second, regarding concerns that our newly proposed metric has not been sufficiently compared with previous works on training instability, we would like to note that we include experiments in Figure 2 of our work showing the failure of a widely used loss landscape visualization technique proposed by Li et al. The new indicator aims to find a proxy for training instability, but we are unaware of alternative approaches for detecting training instabilities that have been verified on autoregressive models. However, we can provide further evidence of the efficacy of our method by observing how a model initially trained on low precision decreases in loss landscape sharpness after training is switched to standard BF16. The table below includes results for how the proposed metric’s value decreases as a Llama 7B model initially trained using E8M3 was later trained on BF16. This result supports our claim that the proposed method is a good indicator of hidden instabilities.

We have updated our work to reflect the feedback we have received during the review. The newly uploaded PDF has the updated sections in blue. Also, we have fixed the units for the loss sharpness to training steps, the same as in Table 1 of our work.

| Training Steps | Sharpness |
|---:|---|
|  7K  | 1.35 |
|  8K  | 1.14 |
|  9K  | 0.98 |
|  10K  | 0.90 |
|  11K  | 0.87 |
|  12K  | 0.77 |
|  13K  | 0.71 |
|  14K  | 0.63 |
|  15K  | 0.60 |
|  16K  | 0.59 |
|  17K  | 0.57 |
|  18K  | 0.50 |
|  19K  | 0.48 |
|  20K  | 0.48 |
|  21K  | 0.46 |
|  22K  | 0.44 |
|  23K  | 0.40 |
|  24K  | 0.40 |
|  25K  | 0.37 |
|  26K  | 0.34 |
|  27K  | 0.34 |
|  28K  | 0.34 |
|  29K  | 0.33 |
|  30K  | 0.35 |
|  31K  | 0.32 |
|  32K  | 0.32 |
|  33K  | 0.30 |
|  34K  | 0.29 |
|  35K  | 0.29 |
|  36K  | 0.28 |
|  37K  | 0.27 |
|  38K  | 0.26 |
|  39K  | 0.27 |
|  40K  | 0.27 |
|  41K  | 0.25 |
|  42K  | 0.23 |
|  43K  | 0.23 |
|  44K  | 0.24 |

---

### Meta-Review · Area_Chair_ag37 · 2024-12-18

**Metareview:**

The paper addresses the important problem of training LLMs in reduced-precision floating-point representations. The scope is to find if the available methods for training in FP8 precision are robust enough. In this context, the paper proposes new evaluation strategies for the sharpness of the LLM training loss.

Overall, the paper has merits, but the reviews are mixed about the quality of the work. In particular, the impact of the trained models on real-world NLP benchmarks is not exhaustively assessed. Furthermore, the authors were encouraged to compare to SOTA FT8 training methods. On the other hand, the reviewers pointed out that experiments with larger model sizes are necessary to fully evaluate the impact of the FP8 training. Last but not least, it is not clear whether the divergence in the FP8 training is due to the use of an open-source dataset, which is not supported by prior work (reviewer nxKU).

While I do see a potential impact of this paper on the community, I remain of the opinion that the paper is currently not mature enough for acceptance.

**Additional Comments On Reviewer Discussion:**

While the authors made a good effort to address some of the comments, some concerns persist. In particular, the comments of the reviewer nxKU on the difference to the findings of prior work (https://arxiv.org/pdf/2409.12517) were not fully answered.

---

### Decision · Program_Chairs · 2025-01-22

Reject